

# Influence of proxy data uncertainty on data assimilation for the past climate

Anastasios Matsikaris[1,2], Martin Widmann[1], and Johann Jungclaus[2]

[1]University of Birmingham, Edgbaston, Birmingham B15 2TT, UK
[2]Max-Planck-Institute for Meteorology, Hamburg, Germany

*Correspondence to:* ANASTASIOS MATSIKARIS (axm368@bham.ac.uk)

**Abstract.** Data assimilation (DA) is an emerging topic in palaeoclimatology and one of the key challenges in this field. Assimilating proxy-based continental mean temperature reconstructions into the MPI-ESM model showed a lack of information propagation to small spatial scales (Matsikaris et al., 2015b). Here, we investigate whether this lack of regional skill is due to the methodology or to errors in the assimilated reconstructions. Error separation is fundamental, as it can lead to improve-
ments in DA methods. We address the question by performing a new set of simulations, using two different sets of target data; the proxy-based PAGES 2K reconstructions (DA-P scheme), and the HadCRUT3v instrumental observations (DA-I scheme). Again, we employ ensemble-member selection DA using the MPI-ESM model, and assimilate NH continental mean temperatures; the simulated period is 1850-1949 AD. Both DA schemes follow the large-scale target and observed climate variations well, but the assimilation of instrumental data improves the performance. This improvement cannot be seen for Asia, where
the limited instrumental coverage leads to errors in the target data and low skill for the DA-I scheme. No skill on small spatial scales is found for either of the two DA schemes, demonstrating that errors in the assimilated data are not the main reason for the unrealistic representation of the regional temperature variability. It can thus be concluded that assimilating continental mean temperatures is not ideal for providing skill on small spatial scales.

## 1 Introduction

Proxy-based palaeoclimatic reconstructions are spatially incomplete and contain large uncertainties caused by non-climatic noise, reconstruction methods, measurement errors, inadequate understanding of the proxy response to environmental variations and other factors (e.g. Jansen et al., 2007; Jones and Mann, 2004). Standard model simulations cannot follow the real internal climate variability, and include systematic biases and errors in the forcings or in the response to them (Fernandez-Donado et al., 2013). Data assimilation (DA) combines empirical information from proxy data with numerical simulations to
obtain the best climate state estimates, aiming to capture both the forced and internal climate variability (e.g. Widmann et al., 2010; Hakim et al., 2013). This study employs DA using ensemble member selection, a method that has been pioneered by Goosse et al. (2006) (see also Crespin et al., 2009; Goosse et al., 2010, 2012; Annan and Hargreaves, 2012; Matsikaris et al., 2015a). Other approaches to DA have also been employed with success (e.g. Huntley and Hakim, 2010; Pendergrass et al., 2012; Bhend et al., 2012; Steiger et al., 2014), all corroborating that DA has the potential to provide improved reconstructions



for the climate of the past and help increase the understanding of climate variability.

As at small spatial scales, random internal variability plays a dominant role compared to forced climate variability, good skill at these scales cannot be obtained in a forced, free running simulation. In order to realistically simulate regional-scale variability, some main modes of atmospheric circulation need to be constrained by DA. Matsikaris et al. (2015b) assimilated continental mean temperatures for the Northern Hemisphere (NH), and showed that although DA simulations generally follow the large-scale target temperatures well, there is a lack of information propagation to smaller spatial scales. A potential reason for this lack of skill was that continental temperatures might not be ideal for constraining the atmospheric circulation and thus regional temperatures. Two main sources of error are present in this DA setup. The first one is the methodology error, caused by: the use of continental-scale, decadal mean temperatures; limited ensemble size; a RMS error-based cost function; and the selection of only the best member. The second one is the data uncertainty error, i.e. differences between reconstructed and true temperatures in the assimilated data. As Matsikaris et al. (2015b) assimilated proxy-based temperatures, which may have substantial errors, it is an open question whether the lack of added value in small spatial scales is mainly due to the method or also to the errors in the assimilated data.

A fundamental problem in the validation of DA simulations for the past is that the true climate is unknown. This problem can in principle be addressed by simulations for the instrumental period, which allow evaluating the assimilation output against direct observations. In this study, to distinguish between methodology and data error in the DA setup of Matsikaris et al. (2015b), we assimilate instrumental observations (Brohan et al., 2006) and proxy-based reconstructions (PAGES 2K Consortium, 2013) for the period 1850-1949 AD using two different DA schemes. The two sets of DA simulations are referred to as DA-I (for the instrumental-based scheme) and DA-P (for the proxy-based scheme) respectively. Assuming that the instrumental observations constitute perfect input data for DA, the errors can be separated: the error in the DA-I scheme is mainly the methodology error, while the difference in the errors of the two schemes is the consequence of the errors in the proxy-based temperature reconstructions. Error separation is important, as it can lead to improvements of the method. The approach remains however challenging, as the global observational coverage is very incomplete early in the record, leading to an additional source of sampling uncertainty when estimating regional averages, which affects DA. The coverage is not complete even in the more recent years that have the most observations (Brohan et al., 2006), hence the gridded instrumental data cannot be considered perfect.

The aim of this study is twofold: firstly, to evaluate the DA scheme of Matsikaris et al. (2015b) against gridded instrumental observations; and secondly, to investigate the effect of the proxy error on the performance and the lack of small spatial-scale skill in the Matsikaris et al. (2015b) setup, and examine the consequences of unrealistic targets for DA. We use the Earth System Model developed at the Max Planck Institute for Meteorology (MPI-ESM) and simulate 20 ensemble members with each DA scheme. The member selection is performed based on decadal mean temperatures for the NH continents. The ensembles are generated sequentially for sub-periods based on the best members of previous sub-periods. We evaluate the performance





of the two schemes against the target and the observed temperatures for large spatial scales, and compare the obtained cost functions. Next, we examine the skill of the two schemes with respect to the observed regional temperature variability, and interpret the main findings.

## 2  Model, data and method

The study is based on a coarse-resolution version of the MPI-ESM coupled model, which follows the configuration for palaeo-applications (MPIESM-P) described in Jungclaus et al. (2014). The atmosphere model ECHAM6 (Stevens et al., 2013) is run at T31 horizontal resolution ($3.75° \times 3.75°$) with 31 vertical levels. The ocean-sea ice model MPIOM (Marsland et al., 2003) is run at a horizontal resolution of $3.0°$ (GR30) and 40 vertical levels. The model includes the OASIS3 coupler and the land surface model JSBACH (Raddatz et al., 2007). Two sets of DA simulations are conducted, selecting the best among 20 ensemble members, in the first case based upon comparison with the Past Global Changes (PAGES) "2k Network" proxy-based reconstructions (DA-P scheme), and in the second case upon comparison with the HadCRUT3v instrumental observations (DA-I scheme). The PAGES 2K network is a set of continental-scale temperature reconstructions with annual resolution, apart from North America, for which 10- and 30-year averages are provided (PAGES 2K Consortium, 2013). Only the NH continental reconstructions have been assimilated, which include annual mean temperatures for the Arctic (60-90N, 180W-180E), summer means (JJA) for Asia (23.5-55N, 60-160E) and Europe (35-70N, 10W-40E) and decadal means representing all seasons for North America (30-55N, 130-75W). The HadCRUT3v data set provides monthly instrumental surface temperature observations interpolated on a $5° \times 5°$ grid (Brohan et al., 2006). In the assimilation of the HadCRUT3v temperatures, we have used the same continental-scale regions and seasons as the ones defined by the PAGES 2K groups.

The assimilation method follows the ensemble-based DA schemes for degenerate particle filters (e.g. Goosse et al., 2006; Crespin et al., 2009). The implementation of the method is the same as in Matsikaris et al. (2015b). The initial condition of the two ensembles was taken as the last day of 1849 AD from a transient forced simulation starting in 850 AD. Twenty ensemble members were generated in each scheme by introducing small perturbations in an atmospheric diffusion parameter, and after 10 years of simulations, a root mean square (RMS) error-based cost function was used to compare the simulated decadal mean temperatures of the NH continents with the PAGES 2K and HadCRUT3v reconstructions respectively. The member that minimized the cost function was selected as the "analysis" for that period and was used as the initial condition for the subsequent 10-year simulation. The procedure was repeated sequentially until 1950 AD. Before being used in the cost function, the simulations, proxy-based reconstructions and instrumental data were all standardized in order to remove biases in the mean and variance, and to give equal weight to each continent regardless of the variance of the respective temperature time series. The standardization in the DA-P scheme used the long-term 850-1849 AD mean and variance, while for the DA-I



scheme the reference period was 1850-1949 AD. The cost function in the two DA schemes is:

$$CF(t) = \sqrt{\sum_{i=1}^{4} \left(T_{mod}^i(t) - T_{tar}^i(t)\right)^2} \tag{1}$$

where $i$ denotes the NH continents, $T_{mod}^i(t)$ is the standardized modelled decadal mean of the temperatures in each continent and $T_{tar}^i(t)$ is the respective standardized target mean temperature (proxy-based in the DA-P scheme and instrumental-based in the DA-I scheme).

## 3 Results

We address two questions when evaluating the performance of the two DA schemes: i) how well the DA analyses follow the assimilated target; and ii) how well they follow the reality as given in the instrumental data set. The study employs two different metrics. The first one, correlation, measures the similarity of the temperature variations regardless of biases or scaling differences. Significant positive correlations are an indication of a common response to changes in external forcings or a similar realisation of internal variability. The second metric, RMS error, tests how close the DA analysis and the reconstructed or observed temperatures are. It is worth mentioning that the PAGES 2K reconstructions are calibrated against the instrumental data sets. For "composite plus scale methods" the temporal evolution of the reconstruction is given by the proxy data composite, regardless of the calibration record, which is only used for scaling. The situation is different for regression-based methods, where the parameter fitting leads to reconstructions that have a temporal behaviour that is more similar to the instrumental continental temperature series than a simple composite. As the number and quality of proxy records decreases when going back in time, the discrepancies between the real climate and the proxy-based reconstructions can be expected to increase, with the consequence of less realistic teleconnections for the reconstructions. The increase in reconstruction errors for periods further in the past might be larger for regression-based methods in cases where overfitting or temporal instabilities in the regression relationships occur. The skill found in our analysis for the DA-P scheme is thus an upper limit for the skill for periods before 1850 AD.

### 3.1 Consistency with the assimilated target data

For each DA scheme, the simulated NH continental temperatures are first compared with the assimilated target data, i.e. the PAGES 2K proxy-based reconstructions for the case of the DA-P scheme and the HadCRUT3v instrumental data for the case of the DA-I scheme. This evaluates the coherence of the DA analyses with the assimilated target. We compare the consistency for the NH continents and the NH mean using both the standardized time series, as this is the form of the data included in the cost function, and the raw (non-standardized) temperature anomalies. Plots are only shown for the raw anomalies; this is the standard way of presenting simulation output. The comparison is based on decadal means, which is the timescale used in the assimilation. Agreement on smaller timescales can possibly arise due to the common response of simulations and reconstruc-



tions (or observations) to the forcings, but it cannot stem directly from the assimilation (apart from some correlation inherited from the decadal timescale).

Figure 1 compares the NH continental decadal mean temperature anomalies for the DA-P analysis with the PAGES 2K reconstructions (anomalies w.r.t. the 850-1850 AD mean), as well as the the DA-I analysis with the HadCRUT3v reconstruction (anomalies w.r.t. the 1850-1949 AD mean). The DA-P analysis follows the temperature variations of the assimilated proxy-based reconstructions very closely, showing that the DA method is skilful. The correlations between the DA-P analysis and the PAGES 2K reconstructions are 0.93 for the Arctic, 0.86 for Asia, 0.85 for Europe and 0.91 for North America. As correlations are independent of scaling, they are the same for the standardized and the raw temperature anomalies. The DA-I analysis and the instrumental data, HadCRUT3v, are also in good agreement, having significant positive correlations on the decadal timescale for the Arctic (0.96), Europe (0.85) and North America (0.88). They are however strongly inconsistent for Asia (correlation is 0.24). The skill of the assimilation on the inter-annual timescale (not shown) is moderate in both schemes (with the exception of Asia in DA-I), largely owing as mentioned above to the response to the forcings.

The RMS errors are calculated from the decadal mean differences between the DA analyses and the target time series for each continent. With the DA-P scheme, the RMS errors of the standardized time series are relatively high (0.89 for the Arctic, 0.49 for Asia, 0.70 for Europe and 0.56 for North America). The RMS errors for the non-standardized anomalies have lower values in all continents (0.20, 0.11, 0.21 and 0.19 respectively). With the DA-I scheme, the decadal RMS errors for the standardized time series are much lower, in all continents except Asia (0.21 for the Arctic, 0.53 for Asia, 0.35 for Europe and 0.27 for North America). The same holds for the non-standardized anomalies (0.13, 0.20, 0.16 and 0.14 respectively). Overall, in three out of the four continents, despite similar correlations, the analysis and its target for the DA-I scheme are closer than for the DA-P scheme. It therefore appears that DA follows the target more closely when assimilating instrumental data.

The large discrepancy between the DA-I analysis and the instrumental data for Asia is owing to the fact that the mean temperature for the continent in the instrumental data set is not representative of the region due to the sparse observations, as discussed below. The agreement between the instrumental time series and the PAGES 2K series is also much lower than for the other continents. A visual comparison of the temperature variations in the DA-I analysis with the instrumental data for Asia shows that they are in better agreement towards the end of the simulation period, when more observations are available.

## 3.2 Impact of the low observational coverage

The incomplete coverage of the PAGES 2K continental areas by the HadCRUT3v grid cells leads to a spatial sampling error in the continental mean temperatures, in addition to the error due to the uncertainty in the individual grid cells. This spatial sampling error depends on the number and position of the grid cells in the continental region, and increases with the lower observational coverage. Such a case is the region of Asia, where the coverage is extremely sparse at the beginning of the period



and progressively becomes more complete (figure 2). The expected error is exacerbated by the uneven spatial resolution during the first few decades, when coverage was concentrated in the western Pacific Ocean and East China Sea. To check the impact of the poor observational coverage on the mean temperature of Asia, we subsampled the data of the last four decades of the simulation period (1910-1949 AD) to have the same coverage as the first decade (1850-59 AD), and calculated the difference

between the complete and the subsampled average temperatures. The four "low data coverage" means are compared with the respective higher data coverage means in table 1. The differences are on the order of 0.1°C, which is similar to the standard deviation of the DA-I analysis in Asia (0.12, figure 1), indicating a significant impact of the poor observational coverage.

The sampling error in the gridded instrumental data leads to possible inconsistencies with reality and with the forcings, in-
cluding errors in the dynamical links between different continents (e.g. correlations). The limited instrumental coverage is thus expected to lead to errors in the target data for Asia, which can explain the low skill of the DA-I scheme for this continent. As the gridded observational record is an imperfect input data set for our assimilation scheme, the difference between the skill of the assimilation of proxy-based and instrumental reconstructions is not identical to the additional error caused by the proxies. Our initial objective of obtaining a clear error separation can therefore not be fully accomplished.

### 3.3  Cost functions comparison

Another way of measuring the closeness between the simulated and assimilated time series in the two DA schemes is the costfunction, which is the RMS error-based measure that has been used to select the best ensemble member for each decade. The cost functions (shown in figure 3) are based on decadal mean standardized anomalies for the four NH continents, using the
mean and variance of 850-1849 AD in the DA-P scheme and 1850-1949 AD in the DA-I scheme. The best cost functions of the DA-I scheme are consistently lower than the DA-P ones in all decades, in agreement with the lower RMS errors found for the DA-I scheme. The mean of the cost functions for the closest member of the DA-P scheme is 1.27, whereas it is 0.61 for the DA-I scheme. Consistent with the results found in section 3.1, this suggests that less realistic targets, such as noisy proxy-based reconstructions, cannot be followed as well by the model as more realistic ones, such as the instrumental data, especially in a
small ensemble.

Our cost function does not include weights that account for the different levels of uncertainty in the assimilated data or for the different size of the regions represented. For the case of the proxy-based reconstructions, this is mainly because the uncertainty estimates have been calculated by the PAGES 2K groups using different methods for the different regions and thus the errors
are not directly comparable. In addition, the published PAGES 2K uncertainties are defined for data at annual resolution. The decadal means used in our cost function can be expected to have lower uncertainties, but the exact level is difficult to determine due to the autocorrelation of the non-climatic noise in the reconstruction (Moberg and Brattstrom, 2011). For consistency, the instrumental errors (e.g. measurement and bias errors as well as the effect of limited observational coverage on the continental





averages) have not been taken into account either.

## 3.4 Evaluation against the observed climate

The DA-I analysis has already been evaluated against the observed climate, i.e. the instrumental data. We now evaluate the

DA-P analysis against these data, and compare the performance of the two schemes. The reference period for the simulation anomalies is 1961-90 AD, taken from a historical run with the MPI-ESM model, while for the gridded observations, anomalies are calculated w.r.t. the 1961-90 AD mean of the HadCRUT3v data set. The decadal mean temperature anomalies for the DA-P and the DA-I analysis as well as the instrumental data in the NH continents are shown in figure 4. The correlations between the DA-P analysis and the HadCRUT3v data set, given in table 2, are much lower than the ones between the DA-I analysis

and HadCRUT3v in Europe, slightly lower in the Arctic and North America, but higher in Asia. For the NH mean, the DA-I analysis is again more consistent with the observations (table 2).

It may not appear surprising that the correlation of the observed climate with the DA-I analysis is higher than with the DA-P analysis, because the observations have been assimilated in the DA-I scheme. However, this is not trivial as the cost function

was based on RMS-error and not on correlation. A limitation of this part of the analysis is that the HadCRUT3v data set serves a twofold purpose: it is used as both the validation data and the assimilation target for the DA-I scheme. In a hypothetical case where the PAGES 2K reconstructions for the NH mean were closer to reality than the HadCRUT3v mean, the DA-I analysis may still follow the HadCRUT3v reconstruction better. The overall consistency of the DA-P analysis with the instrumental data on the large spatial scales shows that given realistic targets, the Matsikaris et al. (2015b) DA setup, which followed the

same scheme for a previous period (1750-1849 AD), performs well.

The correlations of the decadal mean temperature anomalies between the DA-P analysis and the DA-I one are relatively high in all continents, and modest even in the case of Asia (0.80 for the Arctic, 0.48 for Asia, 0.71 for Europe and 0.78 for North America). The correlation for Asia is lower than the correlations of the other three continents, but it is higher than the

correlation between the DA-I analysis and the HadCRUT3v observations. The assimilation of an unrealistic mean temperature for Asia in the DA-I scheme causes the minimization of the cost function to be dominated by the variations in the other three continents. In general, the DA-I analysis is affected by both the target temperatures and the teleconnections between the continents. As the assimilated mean temperature for the Asia region in the DA-I scheme is unrealistic due to the large sampling error, the influence of the teleconnections leads the DA temperatures being inconsistent with the target data.

## 3.5 Skill on spatial patterns

Simulated temperature variations at regional scale cannot be in agreement with the observed ones in a free running simulation, due to the dominant contribution of random internal variability compared to forced climate variability. Good skill on small spatial structures would therefore be added value and is one of the aims of DA. This skill is not easy to achieve in our setup,



as it relies on constraining large-scale circulation modes by the assimilated continental temperatures. We have examined the spatial maps for each decade for the DA analyses and the observations, for Europe and the NH in winters and summers. The DA patterns for the NH in most decades show no agreement with the observed patterns in either of the two schemes (not shown). The patterns in the European sector (35N-65N, 10W-30E), where the instrumental data set is almost complete, are also inconsistent. As an example, figure 5 shows the European decadal surface air temperature maps for the two DA analyses and the HadCRUT3v gridded observations for the summers of 1850-1859 AD and 1940-1949 AD.

The spatial correlations of the DA analyses, regridded onto the coarser resolution observational grid, with the HadCRUT3v data are not significant. The average decadal correlations for the European summers, winters, NH summers and NH winters are -0.12, -0.14, 0.05 and 0.04 respectively with the DA-P scheme, and 0.24, -0.05, 0.07 and 0.01 with the DA-I scheme. The DA-I analysis has a higher correlation than the DA-P one only for Europe in summer, but this improvement is likely to be happening only by chance, especially as the NH teleconnections in summer are weaker than in winter and the performance of the DA scheme probably not as good. The fact that the assimilation of observational data does not provide any added value on regional scales shows that the errors in the proxy-based reconstructions are not the main source of this lack of skill. The use of the continental scale target temperatures is thus the dominant source of error for the unrealistic representation of the small-scale temperature variability. Other aspects of the methodology, such as the ensemble size, may also have contributions. However, even though larger ensemble sizes might lead to some improvements, the complete lack of skill for our 20-member ensemble indicates that the main reason for the lack of skill is in fact the cost function formulation.

## 4 Summary and discussion

This study has analysed two sets of ensemble simulations, which have assimilated continental-scale decadal instrumental temperature observations and proxy-based reconstructions respectively. On the continental and hemispheric scale, both DA analyses follow the target variations well. However, although the correlations of the two DA analyses with the target temperatures are in general similar, the RMS errors of the DA-I analysis are much lower than those of the DA-P one. An exception for this is the case of Asia, where the DA-I analysis is inconsistent with the observations due to the limited observational coverage, which leads to errors in the target data. Another measure used for evaluating the two DA schemes is the decadal cost functions. The cost functions of the DA-I scheme are found to be consistently lower than those of the DA-P scheme, in agreement with the lower RMS errors of the DA-I analysis. Furthermore, the DA-I analysis outperforms the DA-P one in terms of the correlations against the observed climate (HadCRUT3v data), but not substantially.

The overall performance of the two schemes confirms the expectation that the skill of DA is improved by the use of more realistic assimilation data, as these include more realistic teleconnections. In all continents, the DA temperatures are influenced by both the assimilated data and the teleconnections between the NH continents. In cases where the target is unrealistic, such



as the Asia mean temperature of the DA-I scheme, the influence of the teleconnections leads to the simulated temperatures in the DA analysis being different to the target data. In addition to assessing the differences in the performance of the two DA schemes, the study serves as a validation of the Matsikaris et al. (2015b) DA scheme. In principle, since the DA-P scheme is the same as the Matsikaris et al. (2015b) scheme, which was run for the pre-industrial period (1750-1849 AD), the high

consistency of the DA-P analysis with the observations over the instrumental period on the large spatial scales shows that the Matsikaris et al. (2015b) setup yields analyses that are close to the true continental and hemispheric-scale temperatures given realistic targets. The fact that the HadCRUT3v is used as both the validation data and the assimilation target for the DA-I scheme is a limitation of the study.

No skill on representing the regional temperature variability was found for either of the two DA schemes, as the spatial correlations between both DA analyses and the gridded instrumental data for the NH and the European sectors were non-significant. This demonstrates that even target data with only small errors are associated with low skill on small spatial scales. Therefore, methodology is the main source of this lack of skill. Error separation, which was one of the main objectives of the study, could however only be partly achieved, since the observational record is affected by significant sampling error. The methodology

consists of several components, such as the assimilation of continental means, the ensemble size and the specific formulation of the cost function. Given that the combination of the 20-member ensemble with the cost function formulation leads to a skilful representation of the large-scale variations, we believe that the main reason for the lack of added value on small spatial scales is the use of the continental scale for the assimilated data.

The lack of skill in capturing small-scale temperature variability when assimilating continental mean temperatures might be related to the spatial scale or to the location of the continental-scale target data. A systematic investigation of the optimal spatial scale and optimal locations for assimilated temperatures is essential for the improvement of DA methods for palaeoclimatic applications. If the spatial scale is too large, which is likely to be the case for continental-scale target series, the temperature gradients within the regions that are caused by the major circulation modes are ignored, and the information about the phase of

these modes might thus be lost. At the other end of the spectrum is the assimilation of individual local temperature reconstructions. Although this would in principle capture the full information about spatial temperature fields, the local reconstructions might be associated with much larger errors than reconstructions for larger areas, which are based on a larger number of proxy records. While advanced assimilation methods that take into account the errors of the assimilated data can be expected to work well in this situation, simple methods as those used here might work better if spatially averaged temperatures with smaller

errors are assimilated.

*Acknowledgements.* Support was provided by the Natural Environment Research Council (NERC), the University of Birmingham (UK) and the Max Planck Institute (MPI) for Meteorology in Hamburg (Germany). We sincerely thank Helmuth Haak from MPI Hamburg for his support and guidance on running the model.



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



**Table 1.** Decadal winter mean temperatures (anomalies w.r.t. the 1850-1949 AD mean) for the high and low density coverage of Asia in HadCRUT3v, during the last four decades of the simulation period.

| Decade | 1910-1919 | 1920-1929 | 1930-1939 | 1940-1949 |
|---|---|---|---|---|
| High density mean | 0.01 | -0.10 | -0.01 | 0.11 |
| Low density mean | -0.11 | -0.14 | -0.06 | 0.18 |

**Table 2.** Correlations between the NH continental mean temperatures from the DA analyses and the gridded instrumental data, for the two DA schemes.

| | Arctic | Asia | Europe | N. America | NH mean |
|---|---|---|---|---|---|
| DA-P | 0.90 | 0.62 | 0.45 | 0.84 | 0.78 |
| DA-I | 0.96 | 0.24 | 0.85 | 0.88 | 0.91 |



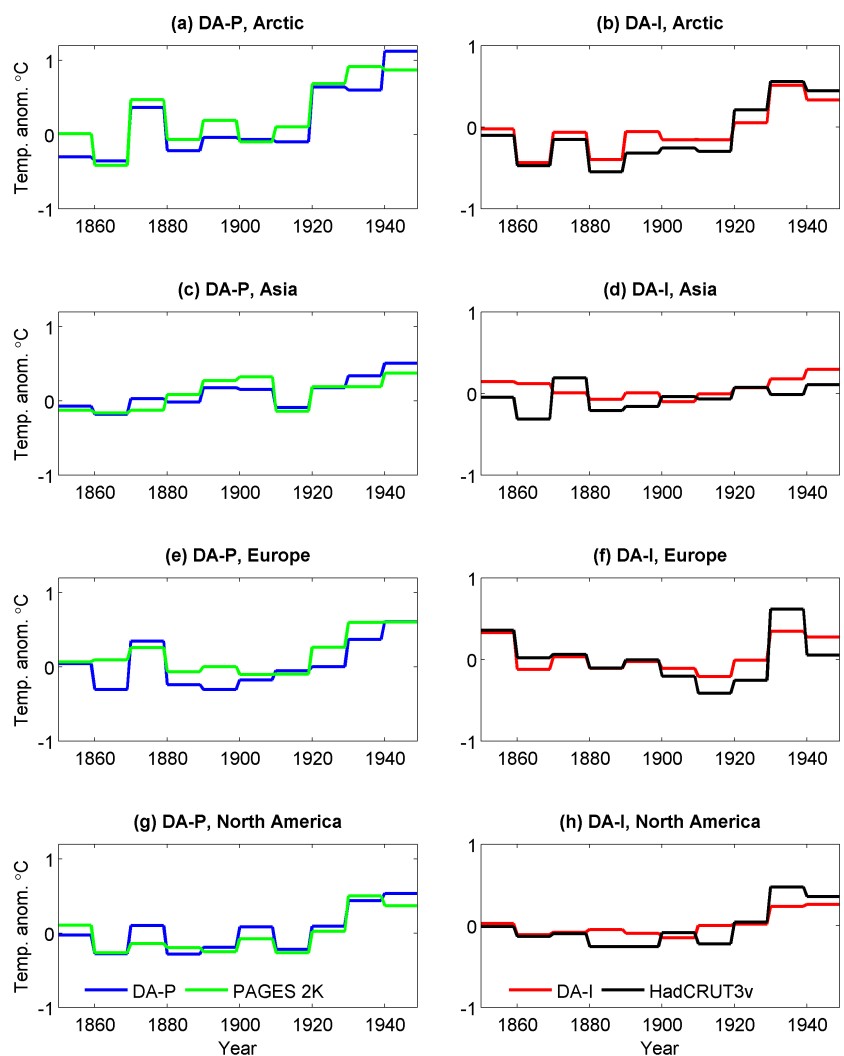

**Figure 1.** Left column: Continental decadal mean temperature anomalies in the NH for the DA-P analysis (blue) and the proxy-based reconstructions (green). The reference period is 850-1849 AD. Right column: Continental decadal mean temperature anomalies in the NH for the DA-I analysis (red) and the instrumental data (black). The reference period is 1850-1949 AD.




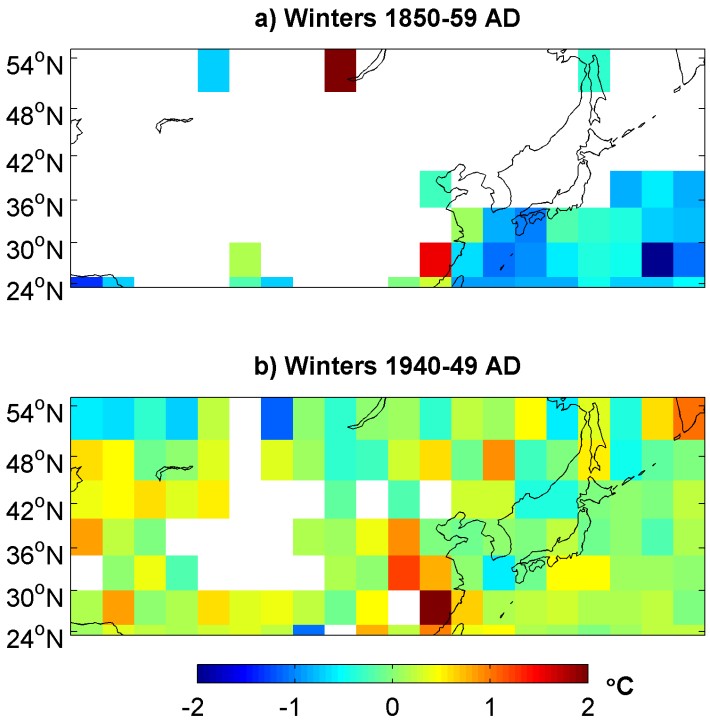

**Figure 2.** HadCRUT3v instrumental coverage in the PAGES 2K box for Asia during the first (1850-1859 AD) and last (1940-1949 AD) decade of the period. The values are decadal winter mean temperatures as anomalies w.r.t. the 1850-1949 AD mean.

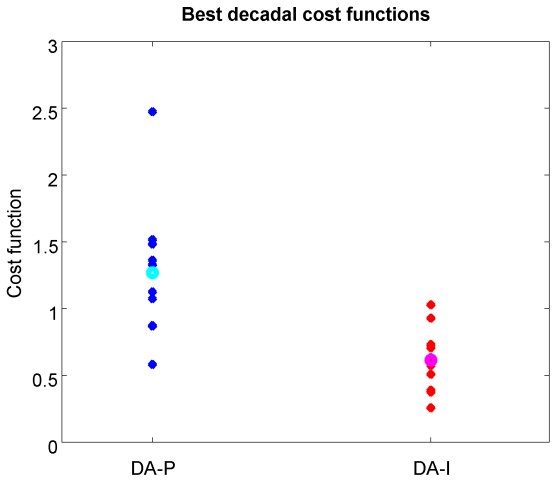

**Figure 3.** Best cost functions for each decade with the DA-P (blue) and DA-I (red) schemes. The mean of the best cost functions for each scheme are also shown (cyan and magenta respectively).




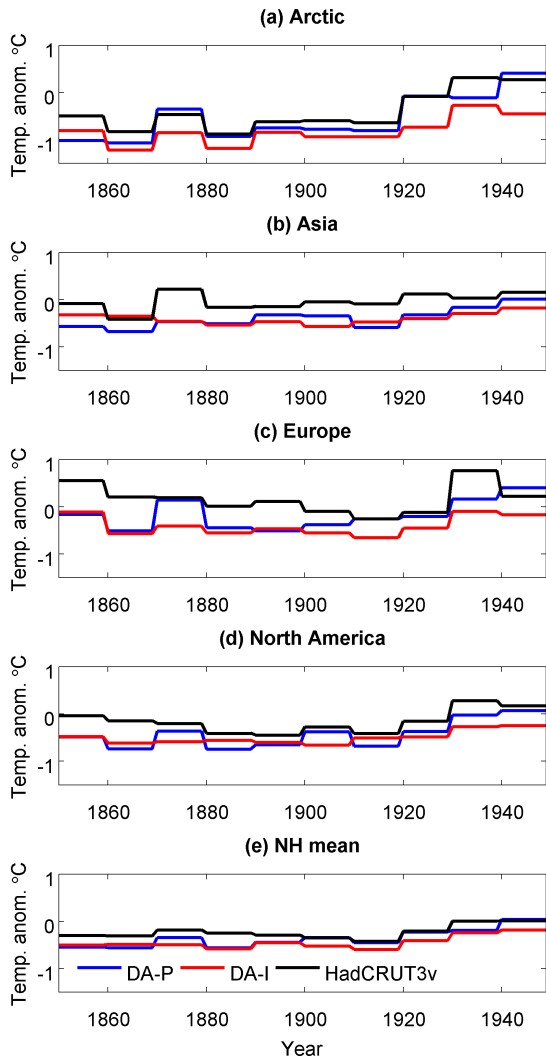

**Figure 4.** NH continental and mean decadal temperature anomalies for the the DA-P analysis (blue), DA-I analysis (red) and the instrumental data (black). The reference period is 1961-1990 AD.





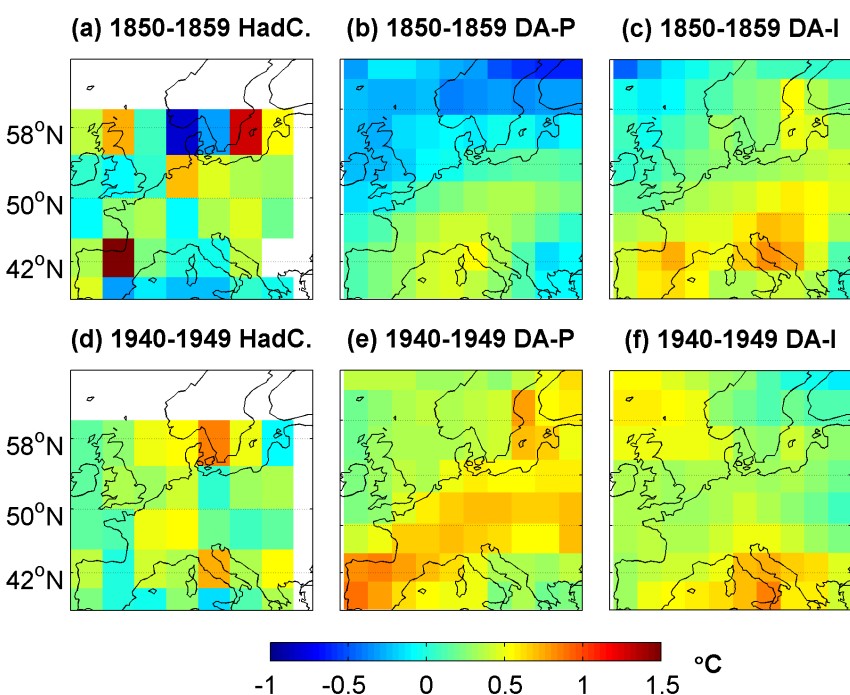

**Figure 5.** European decadal surface air temperatures (anomalies w.r.t. the 1850-1949 AD mean) for the DA-P analysis, the DA-I analysis and the HadCRUT3v reconstructions, for the summers of 1850-1859 AD and 1940-1949 AD.