# Peer review of "Influence of proxy data uncertainty on data assimilation for the past climate"

_Climate of the Past, 2015_

## Referee Comment (RC1) · Anonymous Referee #1 · 19 Feb 2016

The authors aim to investigate reasons for the lack of regional skill which has been often found in experiments when large-scale temperature information is used to constrain or reconstruct temperature fields. Their main conclusion is that this is not due to data errors per se.

I have a major concern with the method being used. Selecting the best simulation out of 20 is not much of a data assimilation method (at least in the case of a high dimensional problem). I see no reason to expect that it will even give repeatable results. That is, a different set of initial random perturbations might give substantially different results, and thus the differences between the two experiments may be an artefact of the specific experiment. I don't see a good way round this other than to repeat the experiments and to see what happens. However, even in the case that the results are repeatable, the method is still very poor and it is debatable whether it really qualifies

as data assimilation.

The authors find no skill on small scales and conclude correctly that data errors were not the cause of this, since this result occurs even when near-perfect data are used. However, while I agree it is certainly true that continental scale temperature does not provide much small scale information, their experiments seem neither necessary nor sufficient to support this conclusion. Not necessary, because of course the information content of large temporal and spatial averages is far too small to distinguish between different climate states, in the same way that describing this page of text as 10% black and 90% white (I'm guessing) cannot tell you what is written on it. Not sufficient, because an equally possible explanation (which is certainly also true) is that even if the data used did have sufficient information, the method applied is almost certainly inadequate to take advantage of this. If the authors doubt my claim, a simple test of this would be to apply the same method while using the whole temperature field data in the assimilation process. That is, pick the ensemble member which fits the whole field best according to their preferred metric, etc. I am confident that such an approach with an ensemble size of 20 - even when the data indisputably does provide all the small scale information - would produce rather poor results, simply because the small ensemble will never contain a simulation which is anywhere near the data. This is essentially the same result that is well known in NWP for simple particle filtering methods (Snyder, C., Bengtsson, T., Bickel, P., & Anderson, J. L. (2008). Obstacles to High-Dimensional Particle Filtering. Monthly Weather Review, 136(12), 4629).

The method used in this manuscript does not, as far as I am aware, have any real justification in terms of solving the Bayesian updating problem implicit to data assimilation. The most that can be said for it is that it selects a simulation which is somewhat closer to the assimilated observations than a randomly-initiated simulation would be. Moreover, in high dimensional applications, the improvement may be very small. Again, the analogy with a page of text may be instructive. If you generate 20 random pages of text and choose the one that has closest to 10% ink, you would not expect the content

to relate any more meaningfully to my review, than the other 19 rejected pages. The most you could say is that it has the same amount of ink.

I have some more minor concerns. For example, the correlations reported for the results might not be as impressive as they appear at first. I suspect that at least part, perhaps a large part, of the correlation is due to the changing forcing applied over the 20th century. You can for instance see what appear to be strong correlations at regional and decadal scale in the famous figure SPM.4 in the 2007 IPCC report, in which simulations there was of course no data assimilation at all, the relationship being purely due to external forcing. In order for your statistics to be informative, it would be necessary to remove the forced element from them. Or perhaps as a simpler alternative, present the correlations achieved by simulations with no data assimilation.

Additionally, some of the comparisons between the methods seem unhelpful. For example, the cost functions differ in their scaling, so cannot be validly used for comparison between the methods (p6 l20). RMS errors could be directly calculated either with no scaling (it might be useful to know how accurately the results match the data, in standard units) or else with a variance scaling that is consistent across the methods. As it stands, there is no way I can easily tell whether the cost of 1.27 for DA-P actually represents a worse model-data mismatch than the 0.61 for DA-I.

―――――――――――――――

---

## Referee Comment (RC2) · Anonymous Referee #2 · 24 Mar 2016

General comments:

A technique for dynamically downscaling continent-scale average temperature is explored using the MPI-ESM model and two sources for the average temperature field: PAGES2K reconstruction and HadCRUT3v. Downscaling proceeds by identifying the ensemble member that best matches the continent-average temperature. While skill is identified for continent-scale temperature, little to no skill is found at smaller scales, consistent with previous studies by the authors. The new finding here is that this lack of skill is not attributable to the source of continent-average temperature. Rather it appears that either continent-scale temperature is not a sufficient constraint on small-scale temperature, or that the model lacks skill on these spatial scales for the timescales considered.

The paper is generally clearly written and makes a specific, if modest, point. It should be publishable with minor revisions.

Specific comments:

p. 4, line 21: I do not understand the point of this sentence.

p. 7, line 28: how do you know this?

p. 8 line 15: An alternative explanation is that weakly constrained large-scale patterns do not constrain small scales.

p. 8, line 18: by this you mean that constraining continent scales is not sufficient?

Figure 3: I don't think a figure is need to convey this information

---

## Author Comment (AC1) · 30 Mar 2016

"The authors aim to investigate reasons for the lack of regional skill which has been often found in experiments when large-scale temperature information is used to constrain or reconstruct temperature fields. Their main conclusion is that this is not due to data errors per se.

I have a major concern with the method being used. Selecting the best simulation out of 20 is not much of a data assimilation method (at least in the case of a high dimensional problem). I see no reason to expect that it will even give repeatable results. That is, a different set of initial random perturbations might give substantially different results, and thus the differences between the two experiments may be an artefact of

the specific experiment. I don't see a good way round this other than to repeat the experiments and to see what happens. However, even in the case that the results are repeatable, the method is still very poor and it is debatable whether it really qualifies as data assimilation."

We thank the reviewer very much for his/her useful and constructive comments. The concerns mentioned by the reviewer are discussed here and addressed in our revised manuscript.

Initially, it is important to note that the data assimilation (DA) problem in palaeoclimatology is different to the respective problem in numerical weather forecasting in many respects. In contrast to the wealth of direct and indirect measurements available for initialising the weather forecasts or for reanalyses, empirical estimates for past climate states are limited and the spatial coverage is incomplete. Moreover, the proxy data constrain seasonal and longer variability rather than individual weather states. In addition, the need for the methods to be efficient enough for long simulations and the high computational cost required introduces another limitation to the use of palaeoclimate DA. In principle, most of the palaeoclimate DA approaches undertaken so far, including the particle filter ensemble selection performed here, are not yet formulated within the standard framework of DA. Other DA schemes, such as Kalman filters, solve the Bayesian updating problem, but are not easy to adapt to palaeoclimate applications because of the assimilation of temporal means rather than individual states. Consequently, we agree with the referee's comment about the DA method not strictly following the classical DA framework, but DA in palaeoclimatology is very ad hoc and pragmatic and indeed far away from 'proper DA'. What we try in our studies is to find how well these alternative DA approaches can work. A wider discussion of these limitations and the fact that the study is only a first step in palaeoclimate DA has been made in the revised introduction.

In terms of how reproducible the results obtained here are, the issue was addressed in the previous paper of the authors (Matsikaris et al, 2015a), who employing the same

methodology showed that the skill of the DA analysis is significantly higher than the skill obtained from random sampling. A resampling method was applied to illustrate the distribution of the skill metrics (correlation and RMS error) when randomly sampling a model as the analysis. In particular, we had calculated the correlations for the Northern Hemisphere (NH) mean temperature for 100 random analyses with the proxy-based reconstructions after randomly selecting one member as the best for each of the 10 decades. The mean correlation of the randomly sampled distribution with the proxy-based reconstruction was 0.48 (with a standard deviation of 0.21), ranging between negative values and 0.8. These correlations were very low compared with the 0.94 correlation found for the DA analysis. The same resampling experiment was performed for the RMS error of the NH mean. The mean RMS error was 0.62 (with a standard deviation of 0.13), ranging between 0.3 and 0.95. The RMS error found for the DA analysis was only 0.11, falling well outside the above range. We therefore believe that the results are robust, at least in terms of the sampling uncertainty. A comment about this has been added in the revised paragraph 2 of Section 3.1.

An additional point regarding the validation of the method is that in contrast to the good skill of the DA schemes in the NH, the agreement between the DA analysis for the Southern Hemisphere and the reconstructions was not good, as expected from the fact that Southern Hemisphere data were not included in the cost function. This was shown in the Matsikaris et al (2015a) study, by both the RMS error and correlation calculations.

"The authors find no skill on small scales and conclude correctly that data errors were not the cause of this, since this result occurs even when near-perfect data are used. However, while I agree it is certainly true that continental scale temperature does not provide much small scale information, their experiments seem neither necessary nor sufficient to support this conclusion. Not necessary, because of course the information content of large temporal and spatial averages is far too small to distinguish between different climate states, in the same way that describing this page of text as 10% black

and 90% white (I'm guessing) cannot tell you what is written on it."

As mentioned in the Matsikaris et al (2015b) paper, upon which the present study is based, even though no information about the local temperatures is assimilated, the assimilation of the NH continental averages might determine to some extent the state of the main modes of circulation variability, such as the Northern Annular Mode (NAM) or the North Atlantic Oscillation (NAO). In principle, prescribing continental temperatures can be expected to constrain the phase of leading circulation modes if these modes have a temperature signal associated with the average continental temperatures. In turn, the temperature signal of these circulation modes can be expected to provide information on temperature variability on sub-continental scales. Furthermore, in that paper, the link between NH continental temperatures and the NH sea level pressure (SLP) field was investigated using 'Maximum Covariance Analysis' and decadal means from the MPI-ESM control simulation. The correlations between the temperature and SLP time expansion coefficients were moderate, indicating the potential for constraining to some extent the large-scale circulation if the continental temperature means are known, which in turn might lead to some skill for small-scale temperature variability in the DA simulations. A comment about this has been added in paragraph 1 of the revised Section 3.5.

"Not sufficient, because an equally possible explanation (which is certainly also true) is that even if the data used did have sufficient information, the method applied is almost certainly inadequate to take advantage of this. If the authors doubt my claim, a simple test of this would be to apply the same method while using the whole temperature field data in the assimilation process. That is, pick the ensemble member which fits the whole field best according to their preferred metric, etc. I am confident that such an approach with an ensemble size of 20 - even when the data indisputably does provide all the small scale information - would produce rather poor results, simply because the small ensemble will never contain a simulation which is anywhere near the data. This is essentially the same result that is well known in NWP for simple particle filtering

methods (Snyder, C., Bengtsson, T., Bickel, P., & Anderson, J. L. (2008). Obstacles to High-Dimensional Particle Filtering. Monthly Weather Review, 136(12), 4629)."

As the reviewer correctly points out, in order to have a good chance of finding a close analogue of an atmospheric state, one requires a large number of ensemble members, if the state space has a high dimension. The high computational cost has however restricted us to running 20 ensemble members for each DA scheme. Nevertheless, in our case, by using the continental averages of decadal mean temperatures as targets for the assimilation process, the dimensionality of the problem is reduced and we have a low number of degrees of freedom for our cost function (less than 3). This makes the finding of a good analogue much more likely with a small ensemble size of 20 members. We also include only the data of the NH in the cost function, in an effort to reduce the degrees of freedom further. The choice of 20 ensemble members is consistent with Bhend et al. (2012), who found that ensembles of size 10 or more can be successful in finding a simulation moderately close to the proxy-based reconstructions, and that considerable skill in regions close to the assimilated data can be found for ensembles of 15 members or more, while larger sizes are needed for areas further away. A relevant comment has been added in the last paragraph of Section 2.

However the dimensionality of the state space that includes the regional temperatures is indeed higher. To systematically address the necessary ensemble size for regional skill given the necessary input information, we plan a study using full field targets if funding allows.

"The method used in this manuscript does not, as far as I am aware, have any real justification in terms of solving the Bayesian updating problem implicit to data assimilation. The most that can be said for it is that it selects a simulation which is somewhat closer to the assimilated observations than a randomly-initiated simulation would be. Moreover, in high dimensional applications, the improvement may be very small. Again, the analogy with a page of text may be instructive. If you generate 20 random pages of text and choose the one that has closest to 10% ink, you would not expect the content

to relate any more meaningfully to my review, than the other 19 rejected pages. The most you could say is that it has the same amount of ink."

Indeed, the method used in this manuscript does not solve the Bayesian updating problem, but is a 'degenerate particle filter' approach. As mentioned previously, DA in palaeoclimatology is very ad hoc and not formulated within the standard DA framework. This is now discussed in the revised introduction.

"I have some more minor concerns. For example, the correlations reported for the results might not be as impressive as they appear at first. I suspect that at least part, perhaps a large part, of the correlation is due to the changing forcing applied over the 20th century. You can for instance see what appear to be strong correlations at regional and decadal scale in the famous figure SPM.4 in the 2007 IPCC report, in which simulations there was of course no data assimilation at all, the relationship being purely due to external forcing. In order for your statistics to be informative, it would be necessary to remove the forced element from them. Or perhaps as a simpler alternative, present the correlations achieved by simulations with no data assimilation."

It is true that a large part of the correlations are due to the changing forcing. This has been one of the main topics of Matsikaris et al. (2015b). The simulation without DA was added in all figures there and the difference of correlation in simulations with and without assimilation was discussed. To keep the current manuscript focused we have not explicitly addressed the issue here again.

In that study, it was shown that the DA scheme gives added value in the correlations of the continental scale, especially on the decadal, but also on shorter timescales. The signal (response to the forcings) to noise (internal variability) ratio increases with the spatial scale. Hemispheric and global means have a relatively small contribution from internal variability, and trying to capture this variability with DA is more difficult, while the dominating forcings are sufficient to provide realistic simulations on these scales. On the continental scale, the influence of the interannual variability is larger, and if a DA

scheme works properly, it can give additional skill compared with a simulation without DA.

It was therefore discussed that under strong forcing, the assimilation scheme cannot offer much added value compared with simulations without assimilation. However, if the forced response in a model is unrealistic, DA can select an ensemble member that, due to internal variability, is closer to reality. Therefore, DA can correct to some extent for unrealistic response to forcing.

"Additionally, some of the comparisons between the methods seem unhelpful. For example, the cost functions differ in their scaling, so cannot be validly used for comparison between the methods (p6 l20). RMS errors could be directly calculated either with no scaling (it might be useful to know how accurately the results match the data, in standard units) or else with a variance scaling that is consistent across the methods. As it stands, there is no way I can easily tell whether the cost of 1.27 for DA-P actually represents a worse model-data mismatch than the 0.61 for DA-I."

We thank the reviewer for spotting this inconsistency. Unscaled cost functions between the two DA schemes cannot be compared due to the very different variance of the datasets. The same goes for the RMS errors; there is no much meaning in comparing raw RMS errors between the two schemes. It is also true that the different periods used in the standardisation in the two schemes makes them again not directly comparable.

However, despite the fact that the standardisation of the DA-I and the DA-P time series is based on different periods, the standardised anomalies are approximately comparable, as the period 1850-1949 AD used in the DA-I scheme is out of the impact of the strong anthropogenic forcing of the late 20th century. The RMS errors and cost functions are consistently lower in the DA-I scheme than the DA-P one. This strongly indicates that there is a stronger match between DA analyses and data in the DA-I scheme. The third paragraph of section 3.1 and the first paragraph of section 3.3 have now been changed to include this discussion. Additionally, Figure 3 was removed from

the revised manuscript as it is not that helpful.  

Anonymous Referee #2

General comments:

"A technique for dynamically downscaling continent-scale average temperature is explored using the MPI-ESM model and two sources for the average temperature field: PAGES2K reconstruction and HadCRUT3v. Downscaling proceeds by identifying the ensemble member that best matches the continent-average temperature. While skill is identified for continent-scale temperature, little to no skill is found at smaller scales, consistent with previous studies by the authors. The new finding here is that this lack of skill is not attributable to the source of continent-average temperature. Rather it appears that either continent-scale temperature is not a sufficient constraint on small-scale temperature, or that the model lacks skill on these spatial scales for the timescales considered. The paper is generally clearly written and makes a specific, if modest, point. It should be publishable with minor revisions."

We are grateful to the reviewer for their useful comments. The points mentioned were addressed in the revised manuscript.

Specific comments:

p. 4, line 21:" I do not understand the point of this sentence."

The sentence was indeed not clearly phrased. It has now been rephrased in the revised manuscript as follows: ''As the errors in reconstructions can be expected to increase back in time, the skill found in our analysis for the DA-P scheme is an upper limit for the skill of periods before 1850 AD".

p. 7, line 28: "how do you know this?"

This is based on the logic that as the cost function has four terms, if three targets are consistent with each other and a similar simulated state is in the ensemble, this one will

be picked. The fourth target, which has large errors, will likely be dynamically incon-sistent with the three other target temperatures, and thus different from the respective simulated continental temperature (which will be dynamically consistent with the three other simulated temperatures).

p. 8 line 15: "An alternative explanation is that weakly constrained large-scale patterns do not constrain small scales. "

The large-scale temperature patterns are realistic as we have assimilated instrumental temperatures. Also, as shown in Matsikaris et al. (2015b) (based on a "maximum covariance analysis" of links between the NH continental temperatures and the NH sea level pressure field), large-scale circulation is moderately well constrained by the continental temperatures. We agree with the reviewer that the resulting errors in the large-scale circulation contribute to the errors in the small-scale temperature variability. However, we note that the errors in the large-scale circulation are a consequence of the assimilation of continental mean temperatures.

p. 8, line 18:" by this you mean that constraining continent scales is not sufficient? "

Yes, this is basically what we meant here; the text has been changed in the revision to make it clearer.

Figure 3: "I don't think a figure is needed to convey this information"

We thank the reviewer for his suggestion; the figure was removed from the revised manuscript and the text has been updated.

References: Matsikaris, A., Widmann, M., and Jungclaus, J.: On-line and off-line data assimilation in palaeoclimatology: a case study, Climate of the Past, 11, 81–93, doi:10.5194/cp-11-81-2015, 2015a.

Matsikaris, A., Widmann, M., and Jungclaus, J.: Assimilating continental mean tem-peratures to reconstruct the climate of the late preindustrial period, Climate Dynamics, doi:10.1007/s00382-015-2785-9, 2015b.